The association between fluid balance and mortality in patients with ARDS was modified by serum potassium levels: a retrospective study

Zhang Zhongheng zh_zhang1984@hotmail.com
Chen Lin
Department of Critical Care Medicine, Jinhua Municipal Central Hospital, Jinhua hospital of Zhejiang University , Zhejiang , PR China
Juan Hsueh-Fen
Electronic publication date: 2015 Feb 10
Publication date: 2015
Volume: 3
Electronic Location ID: e752
Received 2014 Nov 1; Accepted 2015 Jan 17
Copyright: © 2015 Zhang and Chen
Copyright year: 2015
Copyright holder: Zhang and Chen
License: This is an open access article distributed under the terms of the Creative Commons Attribution License, which permits unrestricted use, distribution, reproduction and adaptation in any medium and for any purpose provided that it is properly attributed. For attribution, the original author(s), title, publication source (PeerJ) and either DOI or URL of the article must be cited.
License URL: https://creativecommons.org/licenses/by/4.0/

Keywords: Intensive care unit, Acute respiratory distress syndrome, Mortality, Mean fluid balance, Fractional polynomial

Funding: The authors declare there was no funding for this work.

==============================
Background and Objective. Acute respiratory distress syndrome (ARDS) is characterized by pulmonary edema and may benefit from conservative fluid management. However, conflicting results exist in the literature. The study aimed to investigate the association between mean fluid balance and mortality outcome in ARDS patients who required invasive mechanical ventilation.

Methods. The study was a secondary analysis of a prospectively collected dataset obtained from the NHLBI Biologic Specimen and Data Repository Information Coordinating Center. ARDS patients with invasive mechanical ventilation were eligible. Demographic and laboratory data were extracted from the dataset. Multivariable regression model was built by stepwise selection of covariates. A fractional polynomial approach was used to test the linearity of mean fluid balance in the model. The potential interactions of mean fluid balance with other variables were tested.

Main Results. A total of 282 patients were eligible for the analysis, including 61 non-survivors with a mortality rate of 21.6%. After stepwise regression analysis, mean fluid balance remained to be an independent predictor of death (OR: 1.00057; 95% CI [1.00034–1.00080]). The two-term model obtained using fractional polynomial analysis was not superior to the linear model. There was significant interaction between mean fluid balance and serum potassium levels (p = 0.011). While the risk of death increased with increasing mean fluid balance at potassium levels of 1.9, 2.9 , 3.9 and 4.9 mmol/l, the risk decreased at potassium level of 5.9 mmol/l.

Conclusion. The present study demonstrates that more positive fluid balance in the first 8 days is significantly associated with increased risk of death. However, the relationship between mean fluid balance and mortality can be modified by serum potassium levels. With hyperkalemia, more positive fluid balance is associated with reduced risk of death.

Introduction

Acute respiratory distress syndrome (ARDS) is a leading cause of mortality and morbidity for critically ill patients. The crude incidence of ARDS is reported to be around 80 per 100,000 person-years, with an in-hospital mortality rate of 38.5% (Rubenfeld et al., 2005). However, these figures vary substantially due to different definitions of the syndrome. ARDS is a clinical syndrome with several distinctive features: hypoxemia, pulmonary infiltrates, proteinaceous pulmonary edema, acute onset after known insults, and absence of elevated hydrostatic pressure as the cause (Costa & Amato, 2013; Force et al., 2012). Great advances have been made in the management of ARDS in recent years (Roch, Guervilly & Papazian, 2011), and ventilation strategy is the most extensively studied. Among various ventilation strategies, low-tidal volume ventilation, prone position, and high positive end-expiratory pressure (PEEP) have shown promising results (Santa Cruz et al., 2013). Pharmacological treatment by using beta-agonist for the clearance of alveolar edema has also received attention (Ortiz-Diaz et al., 2013).

Because one of the most important pathological changes of ARDS is proteinaceous fluid accumulation in the interstitial area of the lung, it follows that fluid restriction may be beneficial. Several investigations have been conducted to test this hypothesis, but showed conflicting results. The Fluid and Catheter Treatment (FACT) trial failed to demonstrate any beneficial effect on mortality outcome with conservative fluid administration, but it resulted in more days free from mechanical ventilation and ICU stay (National Heart et al., 2006). On the other hand, some investigators demonstrated that conservative fluid management resulted in long term cognitive impairment (Carlson & Huang, 2013; Mikkelsen et al., 2012). Most studies in the area utilized filling pressure to guide fluid management, and direct daily fluid balance was not systematically analyzed with a rigorous multivariable model. The present study aimed to investigate the association between fluid balance and mortality outcome in ARDS patients by using rigorous model building strategy.

Methods

Setting and study population

The study was a secondary analysis of data from a randomized controlled trial entitled “Randomized, Placebo-controlled Clinical Trial of an Aerosolized b2-Agonist for Treatment of Acute Lung Injury” (NCT 00434993) (National Heart et al., 2011). The dataset for this trial (whose protocol documentation is available at https://biolincc.nhlbi.nih.gov/static/studies/sails/Protocol.pdf) was obtained from National Heart, Lung, Blood Institute (NHLBI) Biologic Specimen and Data Repository Information Coordinating Center (https://biolincc.nhlbi.nih.gov/home/). The study was approved by the ethics committee of Jinhua municipal central hospital (2013-006) and informed consent was obtained in the original clinical trial.

Subjects of the study were recruited from 33 hospitals of the National Heart, Lung, Blood Institute ARDS Clinical Trial Network from August 6, 2007 to July 7, 2008. Patients were deemed eligible if (1) they had bilateral pulmonary infiltrates consistent with edema on chest X-ray; (2) had a ratio of arterial oxygen pressure (PaO2) to oxygen supply (FiO2) of 300 mmHg or less; (3) had no clinical evidence of left atrial hypertension; and (4) they were intubated and mechanically ventilated. Exclusion criteria were patients with chronic lung disease, patients unable to obtain informed consent, time window exceeded, acute myocardial infarction, high 6 month mortality, chronic liver disease, physician refusal, not committed to full support, neuromuscular disease and other unknown reasons (see supplemental material of the original study for more details) (National Heart et al., 2011). Of the 2,688 subjects being screened, 2,406 were excluded due to these reasons. A total of 282 mechanically ventilated patients with ARDS were finally included in the dataset.

The dataset contained the following information: demographics, types of ICU, causes of ARDS, comorbidity score (we assigned one point score for each additional one coexisting disease), lowest mean arterial pressure on admission, laboratory findings on enrollment (sodium, potassium, hemoglobin, glucose, bicarbonate, PaO2/FiO2, pH value and serum creatinine). Urine output, fluid intake and output were reported daily for up to 8 days or until the subject left the study. The dataset recorded comorbidities including chronic dialysis, leukemia, non-Hodgkin’s lymphoma, solid tumor with metastasis, immunosuppression, hepatic failure with coma or encephalopathy, cirrhosis, diabetes mellitus, hypertension, prior myocardial infarction, congestive heart failure, peripheral vascular disease, prior stroke with sequelae, dementia, chronic pulmonary disease, arthritis, peptic ulcer disease.

Patients were followed up for 90 days, and those who survived to 90 days were considered as survivors. Otherwise, they were considered as non-survivors. The secondary outcome was unassisted breathing (UAB), which was defined as time to the first UAB. Subsequent return to mechanical ventilation was not taken into consideration.

Statistical analysis

The distribution of continuous variables was determined by inspection of the histogram and tested based on skewness and kurtosis. Continuous data were expressed as mean and standard deviation and median and interquartile range as appropriate, and their comparisons were made by using the t test or the Wilcoxon rank-sum test. Categorical data such as the types of ICU and causes of ARDS were expressed as the number and percentage, and their differences between survivors and non-survivors were tested by using Pearsons χ2 test.

Mortality was used as a binary outcome, and a logistic regression model was built to adjust for potential confounders. The initial model included all variables with p < 0.3 in bivariate analysis (Table 1). The main effect model was then established by using a stepwise backward elimination approach with the significance levels of removal and addition of 0.2 and 0.1, respectively. An important assumption for the continuous variable (mean fluid balance) in the logistic regression model is that the variable is in a linear relationship with the outcome in the logit scale. If this assumption does not hold true, the fitted model may not reflect the true relationship between mean fluid balance and mortality. We used fractional polynomials to determine whether the curvilinear model was better than the linear model (Royston & Altman, 1994). We first determined the best fitting of one-term and two-term models by choosing power transformations from the set <−2, −1, −0.5, 0, 0.5, 1, 2, 3>, where 0 denoted the log transformation. Then, the best fitting two-term model was compared with linear model. If the two-term model was significantly better than the linear one (p < 0.05), the two-term model was then compared to the best fitting one-term model. Otherwise, the linear model was adopted. The procedure continued until there was no statistical significance and the best fitting model was chosen. This was termed the closed test procedure (Sauerbreia et al., 2006). Interaction terms between mean fluid balance and other variables in the main effect model were tested, and terms with statistical significance (p < 0.05) were included in the model.

Table 1 Demographics and baseline clinical characteristics of ARDS patients by survival status.

Variables	Overall (n = 282)	Survivors (n = 221)	Non-survivors (n = 61)	P value	
Age (years)	51.6 ± 16.2	49.4 ± 16.2	59.8 ± 13.7	<0.001	
Male (n, %)	156 (55.32)	120 (54.30)	36 (59.02)	0.512	
ICU types (n, %)					
Medical ICU	157 (55.67)	121 (54.75)	36 (59.02)	0.553	
Mixed ICU	54 (19.15)	45 (20.36)	9 (14.75)	0.324	
Surgical ICU	31 (10.99)	25 (11.31)	6 (9.84)	0.744	
Othersa	40 (14.18)	30 (13.57)	10 (16.39)	0.576	
Causes of ARDS (n, %)					
Sepsis	156 (55.32)	118 (53.39)	38 (62.30)	0.216	
Transfusion	14 (4.96)	12 (5.43)	2 (3.28)	0.494	
Aspiration	71 (25.18)	54 (24.43)	17 (27.87)	0.584	
Pneumonia	165 (58.51)	130 (58.82)	35 (57.38)	0.839	
Others	24 (8.51)	22 (9.95)	2 (3.28)	0.098	
Comorbidity scoreb	1.59 ± 1.34	1.40 ± 1.28	2.25 ± 1.35	<0.001	
Lowest mean arterial pressure (mmHg)	60.9 ± 11.4	61.5 ± 11.9	58.8 ± 9.3	0.1053	
On enrollment laboratory					
Hemoglobin (g/dl)	10.40 ± 2.22	10.52 ± 2.30	9.95 ± 1.82	0.075	
Sodium (mmol/l)	138.9 ± 5.7	138.9 ± 5.6	138.9 ± 6.4	0.997	
Potassium (mEq/l)	3.96 ± 0.62	3.90 ± 0.58	4.18 ± 0.71	0.0019	
Glucose (mg/dl)	126.4 ± 55.0	128.1 ± 50.4	121.4 ± 69.2	0.402	
Bicarbonate (mEq/l)	22.56 ± 5.50	23.03 ± 5.49	20.84 ± 5.23	0.0056	
PaO2/FiO2 (mmHg)	161.76 ± 78.90	160.30 ± 82.82	166.92 ± 78.90	0.5813	
PaCO2 (mmHg)	40.01 ± 11.83	40.20 ± 10.82	39.35 ± 14.97	0.6232	
pH value	7.35 ± 0.10	7.35 ± 0.10	7.31 ± 0.11	0.0075	
Creatinine (mg/dl)	1.91 ± 1.65	1.85 ± 1.71	2.10 ± 1.43	0.2946	
Urine output day 0 (ml/24h)	1,698 ± 1,413	1,798 ± 1,405	1,339 ± 1,393	0.0244	
Fluid balance day 0 (ml/24h)	2,814 ± 3,590	2,588 ± 3,435	3,628 ± 4,025	0.045	
Cumulative balance in 8 days	5,317 ± 10,952	3,578 ± 9,465	11,614 ± 13,485	<0.001	
Mean fluid balance in 8 days	749 ± 1,601	427 ± 1,179	1,913 ± 2,271	<0.001	
Notes.

a Others include trauma, coronary care unit, burn care unit, cardiac surgery ICU, and neuro ICU.

b Comorbidities include chronic dialysis, leukemia, non-Hodgkin’s lymphoma, solid tumor with metastasis, immunosuppression, hepatic failure with coma or encephalopathy, cirrhosis, diabetes mellitus, hypertension, prior myocardial infarction, congestive heart failure, peripheral vascular disease, prior stroke with sequelae, dementia, chronic pulmonary disease, arthritis, peptic ulcer disease.

ICU Intensive care unit

ARDS Acute respiratory distress syndrome

The Hosmer-Lemeshow goodness-of-fit test was performed to examine the model fit. Another important characteristic of the fitted model is its discrimination power; that is, how accurate the fitted model can predict outcome. We examined this by graphics. The probability of death was depicted on the horizontal axis against the observed outcome in the y-axis. Furthermore, the prediction power was assessed by using the receiver operating characteristics curve (ROC), and the area under the ROC was reported (Royston & Altman, 2010).

All statistical analyses was performed by using Stata 13 (StataCorp, College Station, Texas 77845, USA), and a p < 0.05 was considered to be statistically significant.

Results

The baseline characteristics of included ARDS patients are shown in Table 1. There were 61 non-survivors during the observation period, with a mortality rate of 21.6%. As expected, non-survivors were significantly older than survivors (59.8 ± 13.7 versus 49.4 ± 16.2 years, p < 0.001). There was no difference between survivors and non-survivors in gender, ICU types, causes of ARDS, and the lowest mean blood pressure. However, non-survivors had higher comorbidity burden than survivors (2.25 versus 1.40; p < 0.001). With respect to laboratory measurements on entry, only potassium and bicarbonate were found to be associated with mortality outcome. Other variables were not statistically significant. Survivors showed lower serum potassium levels than non-survivors (3.90 ± 0.58 versus 4.18 ± 0.71 mEq /l; p = 0.0019). Twenty-four h urine output on Day 0 was significantly lower in non-survivors than in survivors (1,339 ± 1,393 versus 1,798 ± 1,405 ml; p = 0.0244). Cumulative fluid balance was significantly higher in non-survivors than in survivors (11,614 ± 13,485 versus 3,578 ± 9,465 ml; p < 0.001), and mean fluid balance was also significantly higher in non-survivors than in survivors (1,913 ± 2,271 versus 427 ± 1,179 ml; p < 0.001). Because there were 26 comparisons between survivors and non-survivors, it was subject to the problem of multiple comparisons. A Bonferroni-adjusted significance level of 0.002 was calculated to account for the increased possibility of type-I error. The mean fluid balance remained statistically significant at this Bonferroni-adjusted significance level. Figure 1 displays the distributions of fluid intake and output from Day 0 to Day 8. More positive fluid balance was shown in the first 3 days, and thereafter the fluid balance was approximately zero. The results of principal component analysis were shown in Fig. 2. The two components were chosen because eigenvalues for the first two principal component (PC) were greater than 1. Biplot shows that the multi-dimensional data were represented by two PCs. Biplot is a visualization technique for investigating the inter-relationships between the observations and variables in multivariate data. The component loading plot showed that PC loadings measure the importance of each variable in accounting for the variability in the PC. PC scores are the derived composite scores computed for each observation based on the eigenvectors for each PC.

Figure 1 Fluid intake and output from Day 0 to Day 8.

More positive fluid balance was shown in the first 3 days, and thereafter the fluid balance was approximately zero.

Figure 2 Principal component analysis (PCA) for the multivariate dataset.

Two components were chosen because eigenvalues for the first two principal component (PC) were greater than 1. Biplot shows the multi-dimensional data were represented by two PCs. Biplot (B) is a visualization technique for investigating the inter-relationships between the observations and variables in multivariate data. The component loading plot showed that PC loadings measure the importance of each variable in accounting for the variability in the PC. PC scores are the derived composite scores computed for each observation based on the eigenvectors for each PC.

After stepwise selection and elimination, five variables remained in the model (main effect model, Table 2): mean fluid balance (OR: 1.77; 95% CI [1.42–2.22]), age (OR: 1.03; 95% CI [1.01–1.06]); potassium (OR: 1.84; 95% CI [1.10–3.06]); hemoglobin (OR: 0.83; 95% CI [0.68–1.00]); and comorbidity (OR: 1.26; 95% CI [0.97–1.64]). In fractional polynomial analysis, the two-term model was significantly better than the model without the variable (mean fluid balance) with a difference of deviance of 34.8 (p < 0.001). However, the two-term model was not significantly better than the linear model (difference of deviance: 2.5; p = 0.111, Table 3), and thus the linear model was adopted for simplicity (Table 4). When interaction terms were entered into the model, we found that the term mean fluid balance × potassium was statistically significant, indicating that the effect of fluid balance on mortality was modified by potassium levels. To make the result more comprehensible to the audience for the subject matter, graphical presentation of the result was made in five potassium levels (Fig. 3). In all four levels, the probability of death increased exponentially with increasing mean fluid balance. Interestingly, in patients with hyperkalemia, the probability of death decreased with increasing mean fluid balance. This final model was well fitted as reflected by a Hosmer-Lemeshow χ2 of 5.26 (p = 0.7292). The model discrimination is graphically shown in Fig. 4. The scatter plot showed that survivors were mostly gathered at the region with lower probability of death, indicating a good negative predictive value of the model. The ROC curve showed that the diagnostic performance of the model was excellent, with an area under ROC of 0.84. Figure 5 displays Kaplan–Meier survivor and failure curves, stratified by median mean fluid balance. Figure 5A shows the probability of survival, and the result indicates that less mean fluid balance is associated with higher survival rate (p = 0.0007 by rog-rank test). In Fig. 5B, less fluid balance is associated with higher rate of returning to UAB (p < 0.001 with log-rank test). The results were robust after adjustment with other confounding factors in Cox proportional hazards model (Tables 5 and 6).

Figure 3 Graphical presentation of the association between mean fluid balance and probability of death, stratified by serum potassium levels.

“S”-shaped relationship between mean fluid balance and risk of mortality was shown for potassium levels at 1.9, 2.9 and 3.9 mmol/l. The relationship was more linear at potassium level of 4.9 mmol/l. Inverse relationship between mean fluid balance and risk of mortality was found at potassium level of 5.9 mmol/l. The relationship was not sensitive to potassium levels (in (B) we set potassium levels at 2, 3, 4, 5 and 6).

Figure 4 Graphical presentation of model discrimination.

The scatter plot (A) showed that survivors were mostly gathered at the region with lower probability of death (left x-axis), indicating a good negative predictive value of the model. The ROC curve (C) showed that the diagnostic performance of the model was excellent, with an area under ROC of 0.84.

Figure 5 Kaplan-Meier survivor and failure curves, stratified by median mean fluid balance.

(A) shows the probability of survival and the result indicates that less mean fluid balance is associated with higher survival rate (p = 0.0007 by rog-rank test). In (B) less fluid balance is associated with higher rate of returning to UAB (p < 0.001 with log-rank test).

Table 2 Main effect model after stepwise selection of covariates.a

	Odds
ratio	Standard
error	Lower limit
of 95% CI	Upper limit
of 95% CI	p	
Mean balance
(with each 1,000 ml increase)	1.77	0.20	1.42	2.22	<0.001	
Age	1.03	0.01	1.01	1.06	0.01	
Comorbidity	1.26	0.17	0.97	1.64	0.09	
Hemoglobin	0.83	0.08	0.68	1.00	0.05	
Potassium	1.84	0.48	1.10	3.06	0.02	
Notes.

a In this multivariable model, mortality was treated as the binary dependent variable. The initial model was built by incorporating all variables with p < 0.3 in bivariate analysis. The main effect model was established by using stepwise backward elimination approach with the significance levels of removal and addition of 0.2 and 0.1, respectively.

Table 3 Comparisons of fractional polynomial models.

The model selection was performed using closed test procedure. By comparing to the model without variable (omitted model), the two-term model (m = 2) was significantly better with a difference of deviance of 34.8. However, the two-term model was not significantly better than the linear model (difference of deviance: 2.5; p = 0.111), and thus the linear model was adopted for simplicity.

Mean balance	Degree of freedom	Deviance	Difference of deviance	p	Powers	
Omitted	0	257.927	34.789	0.000		
Linear	1	225.680	2.542	0.111	1	
m = 1	1	223.775	0.637	0.425	3	
m = 2	2	223.138	0.000	–	3 3	

Table 4 Final model including interaction terms.a

Variables	Odds ratio	95% CI	P value	
Age	1.03	1.007–1.057	0.011	
Potassium	3.18	1.63–6.20	0.001	
Hemoglobin	0.82	0.67–0.99	0.043	
Comorbidity (0 as the reference)				
1	2.22	0.69–7.15	0.180	
2	1.38	0.40–4.83	0.612	
3	3.88	1.16–13.0	0.028	
4	2.96	0.60–14.6	0.182	
5	0.82	0.09–7.54	0.861	
Mean balance	1.003	1.001–1.004	0.001	
Mean balance × potassium	0.9995	0.9991–0.9999	0.011	
Notes.

a All possible interactions between mean balance and other covariates were evaluated and only the term Mean balance × potassium was statistically significant. Goodness-of-fit test showed the Hosmer-Lemeshow χ2 was 5.26 (p = 0.7292).

Table 5 Cox proportional hazards model for mortality.

	Hazard
ratio	Standard
error	Lower limit
of 95% CI	Upper limit
of 95% CI	p	
Mean balance
(with each 1,000 ml increase)	1.58	0.10	1.40	1.79	0.00	
Age	1.02	0.01	1.00	1.03	0.05	
Comorbidity	1.30	0.13	1.06	1.59	0.01	
Hemoglobin	0.92	0.07	0.80	1.06	0.24	
Potassium	1.46	0.24	1.06	2.00	0.02	

Table 6 Cox proportional hazards model for unassisted breathing.

	Hazard
ratio	Standard
error	Lower limit
of 95% CI	Upper limit
of 95% CI	p	
Mean balance
(with each 1,000 ml increase)	0.71	0.04	0.64	0.79	0.00	
Age	1.00	0.00	0.99	1.01	0.98	
Comorbidity	0.91	0.05	0.81	1.02	0.11	
Hemoglobin	1.00	0.03	0.94	1.06	0.92	
Potassium	0.83	0.09	0.67	1.03	0.09	

Discussion

The present study demonstrates that mean fluid balance in the first 8 days is significantly associated with mortality outcome. More positive fluid balance is associated with significantly increased risk of death, and the result is robust after adjustment for shock status, age, comorbidity and plasma hemoglobin. Unexpectedly, the relationship between mean fluid balance and mortality can be modified by serum potassium levels. With hyperkalemia, more positive fluid balance is associated with reduced risk of death. Our study supports previous finding that conservative fluid management would be beneficial for ARDS patients in a short term.

The first investigation into the fluid management in ARDS patients was conducted by Humphrey et al. (1990) two decades ago. However, they did not directly use a fluid restriction strategy as the intervention, but divided patients into low and high pulmonary capillary wedge pressure (PCWP) groups. Although PCWP is correlated well with volume status, they are not a good indicator of fluid responsiveness and can be influenced by multiple confounders. Therefore, the low PCWP may not well represent the optimized volume status, and it is impossible from this study to determine what is the quantity of fluid balance should be prescribed to achieve an optimal outcome. Mitchell et al. (1992) randomized patients with pulmonary artery catheter (PAC) in to PCWP-guided and EVLW-guided groups. They found that EVLW-guided therapy was associated more negative fluid balance and favorable clinical outcomes. However, this study included heterogeneous patients with PAC, and was not specifically designed to explore the effect of fluid therapy in ARDS patients. The milestone study to investigate the effect of fluid restriction on acute lung injury (ALI) was the FACT trial (National Heart et al., 2006), which was a multi-center randomized controlled trial enrolled 1,000 patients with ALI. Conservative and liberal strategies were based on central venous pressure. The conservative group resulted in zero fluid balance in 7 days, whereas the liberal group resulted in +6 l cumulative fluid balance. The conservative strategy improved oxygenation and appeared to reduce the length of the ICU stay, but had no beneficial effect on 60-day mortality. The major difference between our study and the FACT trial is the severity of illness (ARDS requiring invasive mechanical ventilation vs ALI). Most likely, the beneficial effect of fluid restriction can only be detected in ARDS patients in whom the pulmonary edema is more pronounced. In less severe form of lung injury such as ALI, the adverse effect of fluid restriction including tissue hypoperfusion and renal injury is the predominant net effect (Prowle, Kirwan & Bellomo, 2014).

An interesting finding in our study was the interaction between the serum potassium level and mean fluid balance; that is, the effect of the mean fluid balance on mortality was modified by the serum potassium level. In patients with hyperkalemia, mean fluid balance was negatively correlated with the probability of death (Fig. 2). The major cause of hyperkalemia in ICU patients was acute renal injury, and both hyperkalemia and acute renal injury were independent predictors of adverse outcome in critically ill patients (Chertow et al., 2005; Goyal et al., 2012; Herrera-Gutierrez et al., 2013; Lassnigg et al., 2004). Initially, we postulated that the interaction effect was mediated by the kidney. In patients with high risk of renal injury, or those with mild elevation in serum creatinine, electrolyte handling by the kidney is impaired and hyperkalemia may be induced. The progression of renal failure can be reversed with fluid resuscitation (more positive fluid balance) and ensuing improvement in renal perfusion. As a result, more positive fluid balance in patients with hyperkalemia appears to be beneficial. However, there is no sign that serum creatinine was elevated in the non-survivor group in our data, and this postulation remains to be examined. In a retrospective study, McMahon et al. (2012) showed that potassium concentration was a strong independent predictor of all-cause mortality within 30 days after ICU entry, and the effect persisted for one year. It is still unknown whether this association is causal and whether strategies to reduce serum potassium level will improve outcome. Based on current evidence, we proposed that more fluid intake would dilute potassium concentration, and this in turn would translate into improved outcome. Nevertheless, our study is hypothesis-generating at this point, and further investigations are needed to verify this interesting finding.

We employed rigorous methodology in model building and model check. An important concern in our study was that the relationship between mean fluid balance and mortality outcome may not be linear in logit scale. In study design period, we assumed that there would be one critical point above which more positive balance was harmful and below which more positive balance was beneficial. It is a universal phenomenon that biological systems often try to keep variables within a normal range. For instance, our previous study demonstrated that ionized calcium was in a U-shaped relationship with the probability of death in critically ill (Zhang et al., 2014). However, the study failed to identify that the fractional polynomial model was significantly better than the linear model, and thus we can say from our analysis that less fluid balance is better for ARDS patients in the range of −2.5–10 l per day. The study was limited by its observational nature because it was a secondary analysis of a prospectively collected dataset. As described above, only 10% of ARDS patients were included for analysis, and this small proportion may not be representative of the whole ARDS population. The majority of exclusion (28.6%) was of unknown reason or not reported by participating centers. Of course, it is common in randomized controlled trials that only small proportion of patients fulfilled the inclusion/exclusion criteria (Zhang, Ni & Xu, 2014).

The study is limited by its small sample size. The established model may be unstable in future samples and the problem of overfitting may exist (Zhang, 2014b). As a result, our study is hypothesis-generating and requires further confirmation with a larger sample size. One solution to this problem is to use clinical database that is established by using electronic medical record system (Zhang, 2014a). Such a large amount of data is characterized by a large sample size and can support a larger degree of freedom in model building. Furthermore, the problem of multiple comparisons exists in our analysis, which may result in inflated type I error. To address this limitation, we employed a Bonferroni-adjusted significance level to account for the increased possibility of type-I error. At this conservative significance level, the mean fluid balance remained statistically significant.

This manuscript was prepared using Treatment of Acute Lung Injury (ALTA) research material obtained from the National Heart, Lung, Blood Institute (NHLBI) Biologic Specimen and Data Repository Information Coordinating Center and does not necessarily reflect the opinions or views of the National Heart, Lung, Blood Institute.

Additional Information and Declarations

Competing Interests

Author Contributions

The authors declare there are no competing interests.

Zhongheng Zhang conceived and designed the experiments, performed the experiments, analyzed the data, contributed reagents/materials/analysis tools, wrote the paper.

Lin Chen performed the experiments, analyzed the data, contributed reagents/materials/analysis tools, prepared figures and/or tables, reviewed drafts of the paper.

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
