# Peer review of "The association between fluid balance and mortality in patients with ARDS was modified by serum potassium levels: a retrospective study"

_PeerJ, doi:10.7717/peerj.752_

## Round 0.1 · original submission · Major Revisions

This manuscript is suitable to be published in PeerJ, please send it to an English native speaker for editing and submit your revised manuscript according to the reviewers’ comments within 60 days.

Reviewer 1 ·

Basic reporting

Please see the general comments.

Experimental design

Please see the general comments.

Validity of the findings

Please see the general comments.

Additional comments

The manuscript “The association between fluid balance and mortality in patients with ARDS was modified by serum potassium levels: a retrospective study” by Zhang et al. performed a secondary analysis of published ARDS dataset. The authors reported that the mean fluid balance is an independent predictor of death, and it has an interaction effect with the level potassium ions. In general, the study design is straightforward and well-performed. However, there are still some questions/issues that must be clarified before this study can be published. Some specific comments are in the followings.

Specific comments are as follows:
1. The author should send this manuscript for English editing. There are several grammar errors and typos. On page 4 line 106, “who survived to 90 days were considered survivors.” should be “who survived to 90 days were considered as survivors”. On page 4 line 135, the last sentence described “xxx were remained in the model”. However, the verb “remain” cannot be used in the passive sentence. On page 5 line 137, the sentence “Another important characteristic of fitted model is its discrimination power, that is, how accurate can the fitted model predict outcome. No conjunction is used in this sentence and at least a semicolon should be used there. On page 5 line 138, present tense was used but it should be in the past tense. Those errors/typos should be corrected before its publication.
2. Although the p-value of the mean fluid balance is significant, its OR is very close to 1 (1.00057). This value suggests the effect size of the mean fluid balance is very low, which may make it difficult to apply the results into clinics. The authors should address this issue.
3. Please provide a PCA plot of the patients shown in Figure 3A. Such plot may help readers to have a better visualization and follow the idea.
4. The authors should provide p-values in Figure 4 to support their sentences on page 6 lines 185-187. In addition, the authors should perform cox hazard regression model to make the analysis comprehensive.
5. The presentation of “Discussion” should be re-organized. The last paragraphs are totally redundant since they are already described in the first paragraph in “Discussion”.

Reviewer 2 ·

Basic reporting

No Comments

Experimental design

No Comments

Validity of the findings

The findings should be explained carefully.

Additional comments

Manuscript: “The association between fluid balance and mortality in patients with ARDS was modified by serum potassium levels: a retrospective study” by Zhang and Chen.

The authors analyzed a dataset of a randomized controlled trial from the NHLBI Biologic Specimen and Data Repository Information Center. This study aimed to investigate the association between fluid balance and mortality outcome in ARDS patients who required invasive mechanical ventilation. The main conclusions are that mortality outcome is associated with fluid balance and the association can be modified by serum potassium levels.

Major comments
--- Fluid balance was the most important clinical factor discussed in this study. However, the measures of fluid balance seemed not stable. From Table 1, the mean and standard deviation of mean fluid balance (in the first 8 days) in the survivors group, non-survivors group, and overall group were 427±1179, 1913±2271, and 749±1601, respectively. Standard deviations were greater than means obviously especially for the survivors group (the standard deviation was 2.76 folds of the mean)!

--- Three reminders for the authors: (1) Odds ratios of mean fluid balance were very close to 1 in Table 2 and Table 4. (2) The sample size in this study was not so large. 3) Multiple-testing correction was not performed. The impacts of fluid balance on a mortality outcome in ARDS patients should be explained carefully although P values were statistically significant.

Minor essential comments
--- In this paper, the authors dichotomized the continuous survival time into survivors and non-survivors and a logistic regression analysis was performed to examine the association between a mortality outcome and some demographic and clinical variables. This reviewer is curious about: Why the threshold was cut at 90 days? Will the threshold affect the conclusions? Why did not analyze continuous survival time using a survival data analysis directly?

--- The authors should explain why comorbidity was treated as a continuous variable in Table 2 but treated as a categorical variable in Table 4.

--- In Figure 2, will the curves be sensitive to the values of potassium levels? For example, are the curve pattern and conclusion still the same if potassium levels of 2, 3, 4, 5, and 6 are considered? Confidence bands of the five curves should be plotted in Figure 2 and the sample size used for each curve should be provided.

--- Lines 80 – 81: The authors claimed “informed consent was waived due to the nature of the study”. It should be elaborated more why informed consent can be waived in this kind of clinical trial.

Typos:
Line 52: “synfrome” should be corrected and replaced “syndrome”.
Line 114: “standard deviance” should be replaced by “standard deviation”.

---

## Round 0.2 · accepted · Accept

The authors have addressed all the reviewers' comments. I suggest to accept the manuscript.

Reviewer 1 ·

Basic reporting

No comments.

Experimental design

No comments.

Validity of the findings

No comments.

Additional comments

My questions/issues have been addressed.

Reviewer 2 ·

Basic reporting

No Comments

Experimental design

No Comments

Validity of the findings

No Comments

Additional comments

The authors have addressed my previous comments.